# DIVIDE-AND-CONQUER REINFORCEMENT LEARNING

**Dibya Ghosh**[1], **Avi Singh**[1], **Aravind Rajeswaran**[2], **Vikash Kumar**[2], **Sergey Levine**[1]

[1] University of California Berkeley    [2] University of Washington Seattle

`dibya@berkeley.edu`, `avisingh@cs.berkeley.edu`,

`{aravraj, vikash}@cs.washington.edu`, `svlevine@eecs.berkeley.edu`

## ABSTRACT

Standard model-free deep reinforcement learning (RL) algorithms sample a new initial state for each trial, allowing them to optimize policies that can perform well even in highly stochastic environments. However, problems that exhibit considerable initial state variation typically produce high-variance gradient estimates for model-free RL, making direct policy or value function optimization challenging. In this paper, we develop a novel algorithm that instead partitions the initial state space into "slices", and optimizes an ensemble of policies, each on a different slice. The ensemble is gradually unified into a single policy that can succeed on the whole state space. This approach, which we term *divide-and-conquer RL*, is able to solve complex tasks where conventional deep RL methods are ineffective. Our results show that divide-and-conquer RL greatly outperforms conventional policy gradient methods on challenging grasping, manipulation, and locomotion tasks, and exceeds the performance of a variety of prior methods. Videos of policies learned by our algorithm can be viewed at https://sites.google.com/view/dnc-rl/.

## 1 INTRODUCTION

Deep reinforcement learning (RL) algorithms have demonstrated an impressive potential for tackling a wide range of complex tasks, from game playing (Mnih et al., 2015) to robotic manipulation (Levine et al., 2016; Kumar et al., 2016; Popov et al., 2017; Andrychowicz et al., 2017). However, many of the standard benchmark tasks in reinforcement learning, including the Atari benchmark suite (Mnih et al., 2013) and all of the OpenAI gym continuous control benchmarks (Brockman et al., 2016) lack the kind of diversity that is present in realistic environments.

One of the most compelling use cases for RL algorithms is to create autonomous agents that can interact intelligently with diverse stochastic environments. However, such environments present a major challenge for current RL algorithms. Environments which require a lot of diversity can be expressed in the framework of RL as having a "wide" stochastic initial state distribution for the underlying Markov decision process. Highly stochastic initial state distributions lead to high-variance policy gradient estimates, which in turn hamper effective learning. Similarly, diversity and variability can also be incorporated by picking a wide distribution over goals.

In this paper, we explore RL algorithms that are especially well-suited for tasks with a high degree of variability in both initial and goal states. We argue that a large class of practically interesting real-world problems fall into this category, but current RL algorithms are poorly equipped to handle them, as illustrated in our experimental evaluation. Our main observation is that, for tasks with a high degree of initial state variability, it is often much easier to obtain effective solutions to individual parts of the initial state space and then merge these solutions into a single policy, than to solve the entire task as a monolithic stochastic MDP. To that end, we can autonomously partition the state distribution into a set of distinct "slices," and train a separate policy for each slice. For example, if we imagine the task of picking up a block with a robotic arm, different slices might correspond to different initial positions of the block. Similarly, for placing the block, different slices will correspond to the different goal positions. For each slice, the algorithm might train a different policy with a distinct strategy. As the training proceeds, we can gradually merge the distinct policies

into a single global policy that succeeds in the entire space, by employing a combination of mutual KL-divergence constraints and supervised distillation.

It may at first seem surprising that this procedure provides benefit. After all, if the final global policy can solve the entire task, then surely each local policy also has the representational capacity to capture a strategy that is effective on the entire initial state space. However, it is worth considering that a policy in a reinforcement learning algorithm must be able to represent not only the final optimal policy, but also all of the intermediate policies during learning. By decomposing these intermediate policies over the different slices of the initial state space, our method enables effective learning even on tasks with very diverse initial state and goal distributions. Since variation in the initial state distribution leads to high variance gradient estimates, this strategy also benefits from the fact that gradients can be better estimated in the local slices leading to accelerated learning. Intermediate supervised distillation steps help share information between the local policies which accelerates learning for slow learning policies, and helps policies avoid local optima.

The main contribution of this paper is a reinforcement learning algorithm specifically designed for tasks with a high degree of diversity and variability. We term this approach as divide-and-conquer (DnC) reinforcement learning. Detailed empirical evaluation on a variety of difficult robotic manipulation and locomotion scenarios reveals that the proposed DnC algorithm substantially improves the performance over prior techniques.

## 2 RELATED WORK

Prior work has addressed reinforcement learning tasks requiring diverse behaviors, both in locomotion (Heess et al., 2017) and manipulation (Osa et al., 2016; Kober et al., 2012; Andrychowicz et al., 2017; Nair et al., 2017; Rajeswaran et al., 2017a). However, these methods typically make a number of simplifications, such as the use of demonstrations to help guide reinforcement learning (Osa et al., 2016; Kober et al., 2012; Nair et al., 2017; Rajeswaran et al., 2017a), or the use of a higher-level action representation, such as Cartesian end-effector control for a robotic arm (Osa et al., 2016; Kober et al., 2012; Andrychowicz et al., 2017; Nair et al., 2017). We show that the proposed DnC approach can solve manipulation tasks such as grasping and catching directly in the low-level torque action space without the need for demonstrations or a high-level action representation.

The selection of benchmark tasks in this work is significantly more complex than those leveraging Cartesian position action spaces in prior work (Andrychowicz et al., 2017; Nair et al., 2017) or the relatively simple picking setup proposed by Popov et al. (2017), which consists of minimal task variation and a variety of additional shaping rewards. In the domain of locomotion, we show that our approach substantially outperforms the direct policy search method proposed by Heess et al. (2017). Curriculum learning has been applied to similar problems in reinforcement learning, with approaches that require the practitioner to design a sequence of progressively harder subsets of the initial state distribution, culminating in the original task (Asada et al., 1996; Karpathy & van de Panne, 2012). Our method allows for arbitrary decompositions, and we further show that DnC can work with automatically generated decompositions without human intervention.

Our method is related to guided policy search (GPS) algorithms (Levine & Koltun, 2013; Mordatch & Todorov, 2014; Levine et al., 2016). These algorithms train several "local" policies by using a trajectory-centric reinforcement learning method, and a single "global" policy, typically represented by a deep neural network, which attempts to mimic the local policies. The local policies are constrained to the global policy, typically via a KL-divergence constraint. Our method also trains local policies, though the local policies are themselves represented by more flexible, nonlinear neural network policies. Use of neural network policies significantly improves the representational power of the individual controllers and facilitates the use of various off-the-shelf reinforcement learning algorithms. Furthermore, we constrain the various policies to one another, rather than to a single central policy, which we find substantially improves performance, as discussed in Section 5.

Along similar lines, Teh et al. (2017) propose an approach similar to GPS for the purpose of transfer learning, where a single policy is trained to mimic the behavior of policies trained in specific domains. Our approach resembles GPS, in that we decompose a single complex task into local pieces, but also resembles Teh et al. (2017), in that we use nonlinear neural network local policies. Although Teh et al. (2017) propose a method intended for transfer learning, it can be adapted to

the setting of stochastic initial states for comparison. We present results demonstrating that our approach substantially outperforms the method of Teh et al. (2017) in this setting.

# 3 PRELIMINARIES

An episodic Markov decision process (MDP) is defined as $\mathbf{M} = (\mathcal{S}, \mathcal{A}, P, r, \rho)$ where $\mathcal{S}, \mathcal{A}$ are continuous sets of states and actions respectively. $P(s', s, a)$ is the transition probability distribution, $r : \mathcal{S} \to \mathbb{R}$ is the reward function, and $\rho : S \to \mathbb{R}_+$ is the initial state distribution. We consider a modified MDP formulation, where the initial state distribution is conditioned on some variable $\omega$, which we refer to as a "context." Formally, $\Omega = (\omega_i)_{i=1}^n$ is a finite set of contexts, and $\rho : \Omega \times S \to \mathbb{R}_+$ is a joint distribution over contexts $\omega$ and initial states $s_0$. One can imagine that sampling initial states is a two stage process: first, contexts are sampled as $\rho(\omega)$, and then initial states are drawn given the sampled context as $\rho(s|\omega)$. Note that this formulation of a MDP with context is unrelated to the similarly named "contextual MDPs" (Hallak et al., 2015).

For an arbitrary MDP, one can embed the context into the state as an additional state variable with independent distribution structure that factorizes as $P(s_0, \omega) = P(s_0|\omega)P(\omega)$. The context will provide us with a convenient mechanism to solve complex tasks, but we will describe how we can still train policies that, at convergence, no longer require any knowledge of the context, and operate only on the raw state of the MDP. We aim to find a stochastic policy $\pi : \mathcal{S}, \mathcal{A} \to \mathbb{R}_+$ under which the expected reward of the policy $\eta(\pi) = \mathbb{E}_{\tau \sim \pi}[r(\tau)]$ is maximized.

# 4 DIVIDE-AND-CONQUER REINFORCEMENT LEARNING

In this section, we derive our divide-and-conquer reinforcement learning algorithm. We first motivate the approach by describing a policy learning framework for the MDP with context described above, and then introduce a practical algorithm that can implement this framework for complex reinforcement learning problems.

## 4.1 LEARNING POLICIES FOR MDPS WITH CONTEXT

We consider two extensions of the MDP $\mathbf{M}$ that exploit this contextual structure. First, we define an augmented MDP $\mathbf{M}'$ that augments each state with information about the context $(\mathcal{S} \times \Omega, \mathcal{A}, P, r, \rho)$; a trajectory in this MDP is $\tau = ((\omega, s_0), a_0, (\omega, s_1), a_1, \dots)$. We also consider the class of context-restricted MDPs: for a context $\omega$, we have $\mathbf{M}_\omega = (\mathcal{S}, \mathcal{A}, P, r, \rho_\omega)$, where $\rho_\omega(s) = \mathbb{P}(s|\Omega = \omega)$; i.e. the context is always fixed to $\omega$ .

A stochastic policy $\pi$ in the augmented MDP $\mathbf{M}'$ decouples into a family of simpler stochastic policies $\pi = (\pi_i)_{i=1}^n$, where $\pi_i : \mathcal{S}, \mathcal{A} \to [0, 1]$, and $\pi_i(s, a) = \pi((\omega_i, s), a)$. We can consider $\pi_i$ to be a policy for the context-restricted MDP $\mathbf{M}_{\omega_i}$, resulting in an equivalence between optimal policies in augmented MDPs and context-restricted MDPs. A family of optimal policies in the class of context-restricted MDPs is an optimal policy $\pi$ in $\mathbf{M}'$. This implies that policy search in the augmented MDP reduces to policy search in the context-restricted MDPs.

Given a stochastic policy in the augmented MDP $(\pi_i)_{i=1}^n$, we can induce a stochastic policy $\pi_c$ in the original MDP, by defining $\pi_c(s, a) = \sum_{\omega \in \Omega} p(\omega|s)\pi_\omega(s, a)$, where $p(\cdot|s)$ is a belief distribution of what context the trajectory is in. From here on, we refer to $\pi_c$ as the central or global policy, and $\pi_i$ as the context-specific or local policies.

Our insight is that it is important for each local policy to not only be good for its designated context, but also be capable of working in other contexts. Requiring that local policies be capable of working broadly allows for sharing of information so that local policies designated for difficult contexts can bootstrap their solutions off easier contexts. As discussed in the previous section, we seek a policy in the original MDP, and local policies that generalize well to many other contexts induce global policies that are capable of operating in the original MDP, where no context is provided.

In order to find the optimal policy for the original MDP, we search for a policy $\pi = (\pi_i)_{i=1}^n$ in the augmented MDP that maximizes $\eta(\pi) - \alpha \mathbb{E}_\pi[D_{KL}(\pi\|\pi_c)]$ : maximizing expected reward for each instance while remaining close to a central policy, where $\alpha$ is a penalty hyperparameter. This encourages the central policy $\pi_c$ to work for all the contexts, thereby transferring to the original

MDP. Using Jensen's inequality to bound the KL divergence between $\pi$ and $\pi_c$, we minimize the right hand side of Equation 1 as a bound for minimizing the intractable KL divergence optimization problem.

$$\mathbb{E}_\pi[D_{KL}(\pi\|\pi_c)] \leq \sum_{i,j} \rho(\omega_i)\rho(\omega_j)\mathbb{E}_{\pi_i}[D_{KL}(\pi_i\|\pi_j)] \tag{1}$$

Equation 1 shows that finding a set of local policies that translates well into a global policy reduces into minimizing a weighted sum of pairwise KL divergence terms between local policies.

## 4.2 The Divide-and-Conquer Reinforcement Learning Algorithm

We now present a policy optimization algorithm that takes advantage of this contextual starting state, following the framework discussed in the previous section. Given an MDP with structured initial state variation, we add contextual information by inducing contexts from a partition of the initial state distribution. More precisely, for a partition of $\mathcal{S} = \bigcup_{i=1}^n \mathcal{S}_i$, we associate a context $\omega_i$ to each set $\mathcal{S}_i$, so that $\omega = \omega_i$ when $s_0 \in \mathcal{S}_i$. This partition of the initial state space is generated by sampling initial states from the MDP, and running an automated clustering procedure. We use k-means clustering in our evaluation, since we focus on tasks with highly stochastic and structured initial states, and recommend alternative procedures for tasks with more intricate stochasticity.

Having retrieved contexts $(\omega_i)_{i=1}^n$, we search for a global policy $\pi_c$ by learning local policies $(\pi_i)_{i=1}^n$ that maximize expected reward in the individual contexts, while constrained to not diverge from one another. We modify policy gradient algorithms, which directly optimize the parameters of a stochastic policy through local gradient-based methods, to optimize the local policies with our constraints.

In particular, we base our algorithm on trust region policy optimization, TRPO, (Schulman et al., 2015), a policy gradient method which takes gradient steps according to the surrogate loss $\mathcal{L}(\pi)$ in Equation 2 while constraining the mean divergence from the old policy by a fixed constant.

$$\mathcal{L}(\pi) = -\mathbb{E}_{\pi_{old}}\left[A(s,a)\frac{\pi(a|s)}{\pi_{old}(a|s)}\right] \tag{2}$$

We choose TRPO for its practical performance on high-dimensional continuous control problems, but our procedure extends easily to other policy gradient methods (Kakade, 2002; Williams, 1992) as well.

In our framework, we optimize $\eta(\pi) - \alpha\mathbb{E}_\pi[D_{KL}(\pi\|\pi_c)]$, where $\alpha$ determines the relative balancing effect of expected reward and divergence. We adapt the TRPO surrogate loss to this objective, and with the bound in Equation 1, the surrogate objective simplifies to

$$\mathcal{L}(\pi_1\ldots\pi_n) = -\sum_{i=1}^n \mathbb{E}_{\pi_{i,old}}\left[A(s,a)\frac{\pi_i(a|s)}{\pi_{i,old}(a|s)}\right] + \alpha\left(\sum_{i,j}\rho(\omega_i)\rho(\omega_j)\mathbb{E}_{\pi_i}\left[D_{KL}(\pi_i\|\pi_j)\right]\right) \tag{3}$$

The KL divergence penalties encourage each local policy $\pi_i$ to be close to other local policies on its own context $\omega_i$, and to mimic the other local policies on other contexts. As with standard policy gradients, the objective for $\pi_i$ uses trajectories from context $\omega_i$, but the constraint on other contexts adds additional dependencies on trajectories from all the other contexts $(\omega_j)_{j\neq i}$. Despite only taking actions in a restricted context, each local policy is trained with data from the full context distribution. In Equation 4, we consider the loss as a function of a single local policy $\pi_i$, which reveals the dependence on data from the full context distribution, and explicitly lists the pairwise KL divergence penalties.

$$\mathcal{L}(\pi) \propto_{\pi_i} \underbrace{-\mathbb{E}_{\pi_{i,old}}\left[A(s,a)\frac{\pi_i(a|s)}{\pi_{i,old}(a|s)}\right]}_{\text{Maximizes } \eta(\pi_i)} + \alpha\rho(\omega_i)\sum_j \rho(\omega_j)\left(\underbrace{\mathbb{E}_{\pi_i}[D_{KL}(\pi_i\|\pi_j)]}_{\text{Constraint on own context}} + \underbrace{\mathbb{E}_{\pi_j}[D_{KL}(\pi_j\|\pi_i)]}_{\text{Constraint on other contexts}}\right)$$

$$\tag{4}$$

On each iteration, trajectories from each context-restricted MDP $\mathbf{M}_{\omega_i}$ are collected using $\pi_i$, and each local policy $\pi_i$ takes a gradient step with the surrogate loss in succession. It should be noted

that the cost of evaluating and optimizing the KL divergence penalties grows quadratically with the number of contexts, as the number of penalty terms is quadratic. For practical tasks however, the number of contexts will be on the order of 5-10, and the quadratic cost imposes minimal overhead for the TRPO conjugate gradient evaluation.

After repeating the trajectory collection and policy optimization procedure for a fixed number of iterations, we seek to retrieve $\pi_c$, a central policy in the original task from the local policies trained via TRPO. As discussed in Section 4.1, this corresponds to minimizing the KL divergence between $\pi$ and $\pi_c$, which neatly simplifies into a maximum likelihood problem with samples from all the various policies.

$$\mathcal{L}_{center}(\pi_c) = \mathbb{E}_\pi[D_{KL}(\pi(\cdot|s)\|\pi_c(\cdot|s))] \propto \sum_i \rho(\omega_i)\mathbb{E}_{\pi_i}[-\log \pi_c(s,a)] \qquad (5)$$

If $\pi_c$ performs inadequately on the full task, the local policy training procedure is repeated, initializing the local policies to start at $\pi_c$. We alternately optimize the local policies and the global policy in this manner, until convergence. The algorithm is laid out fully in pseudocode below.

---

$R \leftarrow$ Distillation Period
**function** DnC( )
    Sample initial states $s_0$ from the task
    Produce contexts $\omega_1, \omega_2, \ldots \omega_n$ by clustering initial states $s_0$
    Randomly initialize central policy $\pi_c$
    **for** $t = 1, 2 \ldots$ until convergence **do**
        Set $\pi_i = \pi_c$ for all $i = 1 \ldots n$
        **for** $R$ iterations **do**
            Collect trajectories $\mathcal{T}_i$ in context $\omega_i$ using policy $\pi_i$ for all $i = 1 \ldots n$
            **for all** local policies $\pi_i$ **do**
                Take gradient step in surrogate loss $\mathcal{L}$ wrt $\pi_i$
        Minimize $\mathcal{L}_{center}$ w.r.t. $\pi_c$ using previously sampled states $(\mathcal{T}_i)_{i=1}^n$
    **return** $\pi_c$

---

# 5 EXPERIMENTAL EVALUATION

We focus our analysis on tasks spanning two different domains: manipulation and locomotion. Manipulation tasks involve handling an un-actuated object with a robotic arm, and locomotion tasks involve tackling challenging terrains. We illustrate a variety of behaviors in both settings. Standard continuous control benchmarks are known to represent relatively mild representational challenges (Rajeswaran et al., 2017b), and thus it was important to design new tasks that are more challenging in order to illustrate the potential of proposed approach. Tasks were designed to bring out complex contact rich behaviors in settings with considerable variation and diversity. All of our environments are designed and simulated in MuJoCo (Todorov et al., 2012).

Our experiments and analysis aim to address the following questions:

1. Can DnC solve highly complex tasks in a variety of domains, especially tasks that cannot be solved with current conventional policy gradient methods?

2. How does the form of the constraint on the ensemble policies in DnC affect the performance of the final policy, as compared to previously proposed constraints?

We compare **DnC** to the following prior methods and ablated variants:

- **TRPO.** TRPO (Schulman et al., 2015) represents a state-of-the-art policy gradient method, which we use for the standard RL comparison without decomposition into contexts. TRPO is provided with the same batch size as the sum of the batches over all of the policies in our algorithm, to ensure a fair comparison.

- **Distral.** Originally formulated as transfer learning in a discrete action space (Teh et al., 2017), we extend Distral to our stochastic initial state continuous control setting, where each context $\omega$ is a different task. For proper comparison between the algorithms, we port

Distral to the TRPO objective, since empirically TRPO outperforms other policy gradient methods in this domain. This algorithm, which resembles the structure of guided policy search, also trains an ensemble of policies, but constrains them at each gradient step against a single global policy trained with supervised learning, and omits the distillation step that our method performs every $R$ iterations.

- **Unconstrained DnC.** We run the DnC algorithm without any KL constraints. This reduces to running TRPO to train policies $(\pi_i)_{i=1}^n$ on each context, and distilling the resulting local policies every $R$ iterations.

- **Centralized DnC.** Whereas DnC doesn't perform inference on context, centralized DnC uses an oracle to perfectly identify the context $\omega$ from the state $s$. The resulting algorithm is equivalent to the Distral objective, but distills every $R$ steps.

Performance of these methods is highly dependent on the choice of $\alpha$, the penalty hyperparameter that controls how tightly to couple the local policies. For each task, we run a hyperparameter sweep for each method, showing results for the best penalty weight. Furthermore, performance of policy gradient methods like TRPO varies significantly from run to run, so we run each experiment with 5 random seeds, reporting mean statistics and standard deviations. The experimental procedure is detailed more extensively in Appendix A.

For each evaluation task, we use a k-means clustering procedure to partition the initial state space into four contexts, which we found empirically to create stable partitions, and yield high performance across algorithms. We further detail the clustering procedure and examine the effect of partition size on DnC in Appendix C. The focus of our work is finding a single global policy that performs well on the full state space, but we further compare to oracle-based ensemble policies in Appendix D.

## 6 ROBOTIC MANIPULATION

For robotic manipulation, we simulate the Kinova Jaco, a 7 DoF robotic arm with 3 fingers. The agent receives full state information, which includes the current absolute location of external objects such as boxes. The agent uses low-level joint torque control to perform the required actions. Note that use of low-level torque control significantly increases complexity, as raw torque control on a 7 DoF arm requires delicate movements of the joints to perform each task. We describe the tasks below, and present full specifics in Appendix B.

**Picking.** The *Picking* task requires the Jaco to pick up a small block and raise it as high as possible. The agent receives reward only when the block is in the agent's hand. The starting position of the block is randomized within a fixed 30cm by 30cm square surface on the table. Picking up the block from different locations within the workspace require diverse poses, making this task challenging in the torque control framework. TRPO can only solve the picking task from a 4cm by 4cm workspace, and from wider configurations, the Jaco fails to grasp with a high success rate with policies learnt via TRPO.

**Lobbing.** The *Lobbing* task requires the Jaco to flick a block into a target box, which is placed in a randomized location within a 1m by 1m square, far enough that the arm cannot reach it directly. This problem inherits many challenges from the picking task. Furthermore, the sequential nature of grasping and flicking necessitates that information pass temporally and requires synthesis of multiple skills.

**Catching.** In the *Catching* task, a ball is thrown at the robot with randomized initial position and velocity, and the arm must catch it in the air. Fixed reward is awarded every step that the ball is in or next to the hand. This task is particularly challenging due the temporal sensitivity of the problem, since the end-effector needs to be in perfect sync with the flying object successfully finish the grasp. This extreme temporal dependency renders stochastic estimates of the gradients ineffective in guiding the learning.

DnC exceeds the performance of the alternative methods on all of the manipulation tasks, as shown in Figure 1. For each task, we include the average reward, as well as a success rate measure, which provides a more interpretable impression of the performance of the final policy. TRPO by itself is unable to solve any of the tasks, with success rates below 10% in each case. The policies learned by

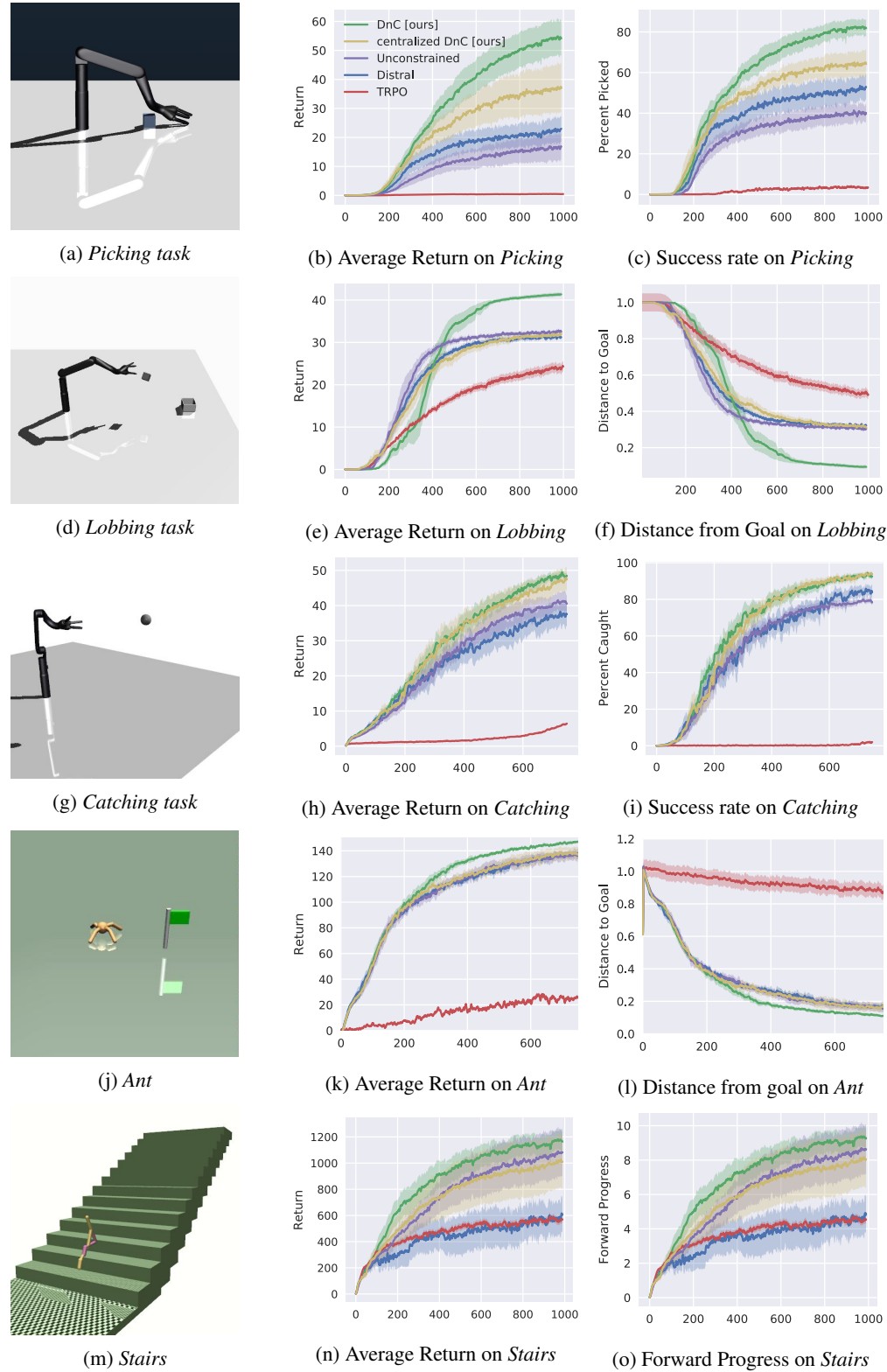

Figure 1: Average return and success rate learning curves of the global policy on Picking, Lobbing, Catching, Ant, and Stairs when partitioned into 4 contexts. Metrics are evaluated each iteration on the global policy distilled from the current local policies at that iteration. On all of the tasks, DnC RL achieves the best results. On the Catching and Ant tasks, DnC performs comparably to the centralized variant, while on the Picking, Lobbing, and Stairs tasks, the full algorithm outperforms all others by a wide margin. All of the experiments are shown with 5 random seeds.

|  | Picker | Lobber | Catcher | Ant Position | Stairs |
|---|---|---|---|---|---|
| TRPO | $0.5 \pm 0.2$ | $23.5 \pm 1.2$ | $6.6 \pm 1.3$ | $23.7 \pm 3.0$ | $563.7 \pm 61.2$ |
| Distral | $23.0 \pm 5.0$ | $31.4 \pm 0.7$ | $36.9 \pm 4.5$ | $137.5 \pm 1.5$ | $616.0 \pm 121.3$ |
| Unconstrained | $16.9 \pm 5.2$ | $32.3 \pm 0.8$ | $40.2 \pm 2.9$ | $138.8 \pm 4.2$ | $1087.0 \pm 196.8$ |
| Centralized DnC (ours) | $37.2 \pm 8.7$ | $31.8 \pm 0.5$ | $\mathbf{46.6 \pm 3.5}$ | $138.6 \pm 4.4$ | $1018.1 \pm 207.1$ |
| DnC (ours) | $\mathbf{55.3 \pm 6.3}$ | $\mathbf{41.3 \pm 0.4}$ | $\mathbf{48.9 \pm 1.0}$ | $\mathbf{146.3 \pm 1.3}$ | $\mathbf{1137.6 \pm 71.5}$ |

Table 1: Overall performance comparison between DnC and competing methods, based on final average return. Performance varies from run to run, so we run each experiment with five random seeds. For each of the tasks, the best performing method is DnC or centralized DnC.

TRPO are qualitatively reasonable, but lack the intricate details required to address the variability of the task.

TRPO fails because of the high stochasticity in the problem and the diversity of optimal behaviour for various initial states, because the algorithm cannot make progress on the full task with such noisy gradients. When we partition the manipulation tasks into contexts, the behavior within each context is much more homogeneous.

Figure 1 shows that DnC outperforms both the adapted Distral variant and the two ablations of our method. On the picking task, DnC has a 16% higher success rate than the next best method, which is an ablated variant of DnC, and on the lobbing task, places the object three times closer to the goal as the other methods do. Both the pairwise KL penalty and the periodic reset in DnC appear to be crucial for the algorithm's performance. In contrast to the methods that share information exclusively though a single global policy, the pairwise KL terms allow for more efficient information exchange. On the Picking task, the centralized variant of DnC struggles to pick up the object pockets along the boundaries of the contexts, likely because the local policies differ too much in these regions, and centralized distillation is insufficient to produce effective behavior.

On the catching task (Figure 1c), the baselines which are distilled every 100 iterations (the DnC variants) all perform well, whereas Distral lags behind. The policy learned by Distral grasps the ball from an awkward orientation, so the grip is unstable and the ball quickly drops out. Since Distral does not distill and reset the local policies, it fails to escape this local optimal behaviour.

## 7 LOCOMOTION

Our locomotion tasks involve learning parameterized navigation skills in two domains.

**Ant Position** In the *Ant Position* task, the quadruped ant is tasked with reaching a particular goal position; the exact goal position is randomly selected along the perimeter of a circle 5m in radius for each trajectory. The ant is penalized for its distance from the goal every timestep. Although moving the ant in a single direction is solved, training an ant to walk to an arbitrary point is difficult because the task is symmetric and the global gradients may be dampened by noise in several directions.

**Stairs** In the *Stairs* task, a planar (2D) bipedal robot must climb a set of stairs, where the stairs have varying heights and lengths. The agent is rewarded for forward progress. Unlike the other tasks in this paper, there exists a single gait that can solve all possible heights, since a policy that can clear the highest stair can also clear lower stairs with no issues. However, optimal behavior that maximizes reward will maintain more specialized gaits for various heights. The agent locally observes the structure of the environment via a perception system that conveys the information about the height of the next step. This task is particularly interesting because it requires the agent to compose and maintain various gaits in an diverse environment with rich contact dynamics.

As in manipulation, we find that DnC performs either on par or better than all of the alternative methods on each task. TRPO is able to solve the Ant task, but requires 400 million samples, whereas variants of our method solve the task in a tenth of the sample complexity. Initial behaviour of TRPO has the ant moving in random directions throughout a trajectory, unable to clearly associate movement in a direction with the goal reward. On the *Stairs* task, TRPO learns to take long striding gaits that perform well on shorter stairs but cause the agent to trip on the taller stairs, because the

reward signal from the shorter stairs is much stronger. In DnC, by separating the gradient updates by context, we can mitigate the effect of a strong reward signal on a context from affecting the policies of the other contexts.

We notice large differences between the gaits learned by the baselines and DnC on the *Stairs* task. DnC learns a striding gait on shorter stairs, and a jumping gait on taller stairs, but it is clearly visible that the two gaits share structure. In contrast, the other partitioning algorithms learn hopping motions that perform well on tall stairs, but are suboptimal on shorter stairs, so brittle to context.

## 8 DISCUSSION AND FUTURE WORK

In this paper, we proposed divide-and-conquer reinforcement learning, an RL algorithm that separates complex tasks into a set of local tasks, each of which can be used to learn a separate policy. These separate policies are constrained against one another to arrive at a single, globally coherent solution, which can then be used to solve the task from any initial state. Our experimental results show that divide-and-conquer reinforcement learning substantially outperforms standard RL algorithms that samples initial and goal states from their respective distributions at each trial, as well as previously proposed methods that employ ensembles of policies. For each of the domains in our experimental evaluation, standard policy gradient methods are generally unable to find a successful solution.

Although our approach improves on the power of standard reinforcement learning methods, it does introduce additional complexity due to the need to train ensembles of policies. Sharing of information across the policies is accomplished by means of KL-divergence constraints, but no other explicit representation sharing is provided. A promising direction for future research is to both reduce the computational burden and improve representation sharing between trained policies with both shared and separate components. Exploring this direction could yield methods that are more efficient both computationally and in terms of experience.

### ACKNOWLEDGEMENTS

This research was supported by the National Science Foundation through IIS-1651843 and IIS-1614653, an ONR Young Investigator Program award, and Berkeley DeepDrive.

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

## A    EXPERIMENTAL DETAILS

To ensure consistency, all the methods tested are implemented on the TRPO objective function, allowing for comparisons between the various types of constraint. In particular, the Distral algorithm is ported from a soft Q-learning setting to TRPO. TRPO was chosen as it outperforms other policy gradient methods on challenging continuous control tasks. To properly compare TRPO to the partition-based methods for sample efficiency, we increase the number of timesteps of simulation used per policy update for TRPO. Explicitly, if $B$ is the number of timesteps simulated used for a single local policy iteration in DnC, and $N$ the number of local policies, then we use $B*N$ timesteps for each policy iteration in TRPO.

Stochastic policies are parametrized as $\pi_\theta(a|s) \sim \mathcal{N}(\mu_\theta(s), \Sigma_\theta)$. The mean,$\mu_\theta(\cdot)$, is a fully-connected neural network with 3 hidden layers containing $150, 100$, and $50$ units respectively. $\Sigma$ is a learned diagonal covariance matrix, and is initially set to $\Sigma = I$.

The primary hyperparameters of concern are the TRPO learning rate $\bar{D}_{KL}$ and the penalty $\alpha$. The TRPO learning rate is global to the task; for each task, to find an appropriate learning rate, we ran TRPO with five learning rates $\{.0025, .005, .01, .02, .04\}$. The penalty parameter is not shared across the methods, since a fixed penalty might yield different magnitudes of constraint for each method. We ran DnC, Centralized DnC, and Distral with five penalty parameters on each task. The penalty parameter with the highest final reward was selected for each algorithm on each task. Because of variance of performance between runs, each experiment was replicated with five random seeds, reporting average and SD statistics.

|  | Picker | Lobber | Catcher | Ant Position | Stairs |
|---|---|---|---|---|---|
| State Space Dimension | 34 | 40 | 34 | 146 | 41 |
| Action Space Dimension | 7 | 7 | 7 | 8 | 6 |
| # Steps per Local Iteration | 30000 | 30000 | 30000 | 50000 | 50000 |
| # Iterations | 1000 | 1000 | 750 | 750 | 1000 |
| Distillation Period | 100 | 100 | 100 | 50 | 100 |
| Learning Rate | .01 | .02 | .02 | .01 | .02 |

## B    TASK DESCRIPTIONS

All the tasks in this work have the agent operate via low-level joint torque control. For Jaco-related tasks, the action space is 7 dimensional, and the control frequency is 20Hz. For target-based tasks, instead of using the true distance to the target, we normalize the distance so the initial distance to the target is 1.

$$\text{normalized distance to target} = \frac{\text{distance to target}}{\text{initial distance to target}}$$

**Picking**    The observation space includes the box position, box velocity, and end-effector position. On each trajectory, the box is placed in an arbitrary location within a 30cm by 30cm square surface of the table.

$$R(s) = \mathbf{1}\{\text{Box in air and Box within 8cm of Jaco end-effector}\}$$

**Lobbing.**    The observation space includes the box position, box velocity, end-effector position, and target position. On each trajectory, the target location is randomized over a 1m by 1m square.

An episode runs until the box is lobbed and lands on the ground. Reward is received only on the final step of the episode when the lobbed box lands; reward is proportional to the box's time in air, $t_{air}$, and the box's normalized distance to target, $d'_{target}$.

$$R(s) = t_{air} + 40 \max(0, 1 - d'_{target})$$

**Catching.**    The observation space includes the ball position, ball velocity, and end-effector position. On each trajectory, both the ball position and velocity are randomized, while ensuring the ball is still "catchable".

$$R(s) = \mathbf{1}\{\text{Ball in air and Ball within 8cm of Jaco end-effector}\}$$

**Ant Position.**    The target location of the ant is chosen randomly on a circle with radius 5m.

The reward function takes into account the normalized distance of the ant to target, $d'_{target}$, and as with the standard quadruped, the magnitude of torque, $\|a\|$, and the magnitude of contact force, $\|c\|$.

$$R(s, a) = 1 - d'_{target} - 0.01\|a\| - 0.001\|c\|$$

**Stairs.**    The planar bipedal robot has a perception system which is used to communicate the local terrain. The height of the platforms are given at 25 points evenly spaced from $0.5$ meters behind the robot to 1 meters in front. On each trajectory, the heights of stairs are randomized between 5 cm and 25 cm, and lengths randomized between 50 cm and 60 cm. The reward weighs the forward velocity $v_x$, and the torque magnitude $\|a\|$.

$$R(s, a) = v_x - 0.5\|a\| + 0.01$$

## C  Automated Partitioning

In this section, we detail the procedure used to partition the initial state space into contexts, and examine performance of DnC as the number of contexts is varied. 10000 initial states are sampled from the task, and are fed through a $K$-means clustering procedure to produce $k$ cluster centers $(c_i)_{i=1}^{k}$. We assign initial states to the context with the closest center:

$$\omega_i = \arg\min_i \|c_i - s_0\|^2$$

The k-means procedure is sensitive to the relative scaling of the state, but we found empirically that the clustering procedure yielded sane partitions on all the benchmark tasks. We examine the performance of DnC with this partitioning scheme when split into two, four, and eight contexts respectively, and for comparison, we also include a manually labelled partition. The manual partition into four contexts is a grid decomposition along the axes of stochasticity. To ensure a fair comparison, the sample complexity is kept constant across variants: when run with two contexts, each local policy consumes twice the number of samples as when run with four contexts.

|                      | Picker        | Lobber       | Catcher      | Ant Position  | Stairs          |
| -------------------- | ------------- | ------------ | ------------ | ------------- | --------------- |
| 2 Contexts           | $14.7 \pm 2.2$ | $42.0 \pm 0.7$ | $39.2 \pm 9.4$ | $145.2 \pm 1.5$ | $1040.8 \pm 44.2$ |
| 4 Contexts           | $55.3 \pm 6.3$ | $41.3 \pm 0.4$ | $48.9 \pm 1.0$ | $146.3 \pm 1.3$ | $1137.6 \pm 71.5$ |
| 4 Contexts (Manual)  | $44.5 \pm 6.8$ | $42.2 \pm 0.7$ | $35.8 \pm 4.1$ | $81.7 \pm 1.8$  | $1218.2 \pm 27.8$ |
| 8 Contexts           | $51.0 \pm 4.3$ | $40.6 \pm 0.6$ | $43.8 \pm 3.7$ | $133.4 \pm 3.3$ | $1084.9 \pm 41.0$ |

On all the tasks, running DnC with four contexts is either the best performing method, or closely matches the best performing method. This indicates a balance between representation sharing within a context, and the benefit from optimizing over small contexts. When run with two contexts, the contexts being optimized over are relatively large, and thus face many of the same issues as TRPO in extracting a signal from a noisy gradient, perhaps best seen in the Picking task. The performance increase from TRPO to two-context DnC however seems to indicate that the distillation and reset of local policies prevents the learning algorithm from being stuck in local optima. When run with eight tasks, we notice a representation sharing issue, since even between very similar initial states, information can only be shared through the KL constraint, which is a bottleneck. This analysis indicates that the choice of the number of clusters is a trade-off between having large enough contexts to share information freely between similar states, and having small enough states to overcome the noise in the policy gradient signal.

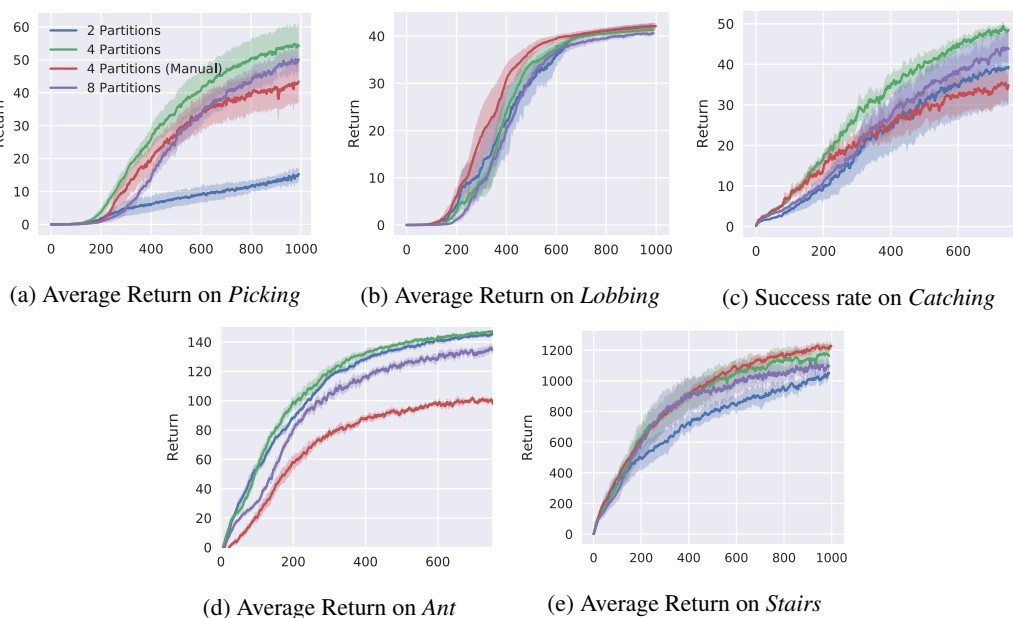

(a) Average Return on *Picking*    (b) Average Return on *Lobbing*    (c) Success rate on *Catching*

(d) Average Return on *Ant*    (e) Average Return on *Stairs*

## D   ORACLE-BASED ABLATIONS

Whereas DnC maintains a global policy to run on all contexts, we consider in this section ablations whose final output is an ensemble of local policies, choosing the appropriate policy on each trajectory via oracle.

**Final Local Policies**   We run the DnC algorithm, and return the ensemble of final local policies instead of the resulting global policy. This method is expected to outperform DnC, since the global policy should be strictly worse than the local ensemble. However, as seen in the table below, the gap in performance is low for the majority of tasks, showing that minimal information is lost in transferring from the ensemble of local policies to the global policy.

|  | Picker | Lobber | Catcher | Ant Position | Stairs |
|---|---|---|---|---|---|
| Final Global Policy (DnC) | $55.3 \pm 6.3$ | $41.3 \pm 0.4$ | $48.9 \pm 1.0$ | $146.3 \pm 1.3$ | $1137.6 \pm 71.5$ |
| Final Local Policies | $56.6 \pm 6.4$ | $41.2 \pm 0.4$ | $50.2 \pm 0.7$ | $146.6 \pm 0.5$ | $1170.0 \pm 68.4$ |

**No Distillation**   We run the DnC algorithm, discarding the distillation step every $R$ iterations. This is equivalent to training local policies with pairwise KL constraints till convergence, and considering the resulting ensemble of local policies. We notice that DnC significantly outperforms the variant without distillation on three of the tasks, and has equivalent performance on the other two. We hypothesize this is because the local policies often become trapped in local minima, and the distillation step helps adjust the policy out of the optima. This is consistent with observations in previous work involving trajectory optimization (Mordatch et al., 2015), where adding a central neural network to which trajectories were distilled significantly increased performance.

|  | Picker | Lobber | Catcher | Ant Position | Stairs |
|---|---|---|---|---|---|
| Distillation (DnC) | $55.3 \pm 6.3$ | $41.3 \pm 0.4$ | $48.9 \pm 1.0$ | $146.3 \pm 1.3$ | $1137.6 \pm 71.5$ |
| No Distillation | $30.2 \pm 6.8$ | $40.6 \pm 1.0$ | $20.3 \pm 2.7$ | $69.8 \pm 1.4$ | $919.5 \pm 28.2$ |

