# OpenReview forum: "Divide-and-Conquer Reinforcement Learning"
_ICLR.cc/2018/Conference — Accept (Poster)_

### Official Review · AnonReviewer1 · 2017-11-27
**Good paper, pushing the limits of RL to harder tasks.**

**Rating:** 7
**Confidence:** 4

**Review:**

This paper presents a reinforcement learning method for learning complex tasks by dividing the state space into slices, learning local policies within each slice, while ensuring that they don't deviate too far from each other, while simultaneously learning a central policy that works across the entire state space in the process. The most closely related works to this one are Guided Policy Search (GPS) and "Distral", and the authors compare and contrast their work with the prior work suitably.

The paper is written well, has good insights, is technically sound, and has all the relevant references. The authors show through several experiments that the divide and conquer (DnC) technique can solve more complex tasks than can be solved with conventional policy gradient methods (TRPO is used as the baseline). The paper and included experiments are a valuable contribution to the community interested in solving harder and harder tasks using reinforcement learning.

For completeness, it would be great to include one more algorithm in the evaluation: an ablation of DnC which does not involve a central policy at all. If the local policies are trained to convergence, (and the context omega is provided by an oracle), how well does this mixture of local policies perform? This result would be instructive to see for each of the tasks.

The partitioning of each task must currently be designed by hand. It would be interesting (in future work) to explore how the partitioning could perhaps be discovered automatically.

---

> ### Author Response · Authors · 2017-12-27
> **Response to Reviewer 1**
>
> Thank you for your very valuable feedback!
>
> We have modified the paper to include comparisons between DnC and two different oracle-based ensembles of local policies in Appendix C.  The first ablation of DnC never distills the policies together, training the local policies to convergence. This ablation performs poorly compared to DnC in most tasks: we hypothesize that the distillation step allows the local policies to escape the local minima that policy gradient methods generally suffer from. Similar observations have been noted in Mordatch et al. [1], where trajectory optimization without distillation to a central neural network underperforms. The other ablation runs DnC, but returns the final local ensemble instead of the final global policy. We observe that this final local ensemble with oracle context performs only marginally better than the final global policy in most tasks, indicating that there is little loss in performance during the distillation process. For both of these variants, the central policy, which must  operate successfully for a wide range of contexts, generalizes better to contexts that are slightly different than the training distribution. Considering that training and testing conditions will almost always differ slightly in practice, even if one has oracle access to the context, it might be beneficial to use the central policy due to its better generalization capability.
>
> Automatic ways to perform the partitioning is indeed an interesting future direction! As a step in this direction, we have updated the paper with a simple automated partitioning scheme in Appendix D. Partitions are automatically generated via a K-means clustering procedure on the initial state distribution to generate contexts, and find that DnC performs well in this case as well. We hope to pursue more elaborate partitioning schemes in future work.
>
> [1] Mordatch et al, Interactive Control of Diverse Complex Characters with Neural Networks, NIPS 2015

---

### Official Review · AnonReviewer2 · 2017-11-27
**Good submission, although could use more evaluation**

**Rating:** 7
**Confidence:** 4

**Review:**

The submission tackles an important problem of learning highly varied skills. The approach relies on dividing the task space into subareas (defined by task context vectors) over which individual policies are trained, but are still required to operate well on tasks outside their context.

The exposition is clear and the method is well-motivated. I see no issues with the mathematical correctness of the claims made in the paper. The experimental results show a convincing benefit over TRPO and Distral on a number of manipulation and locomotion tasks. I would like to have seen more discussion of the computational costs and scaling of the method over TRPO or Distral, as the pairwise KL divergence terms grow quadratically in the number of contexts.

While the method is well-motivated, the division of tasks into subareas seems arbitrarily chosen. It would be very useful for readers to see performance of the algorithm under other task decompositions to alleviate the worries that the algorithm is not sensitive to the decomposition choice.

I would also like to see more discussion of curriculum learning, which also aims at tackling a similar problem of reducing complexity in early stages of training by choosing on simper tasks and progressing to more complex. Would such progressive tasks decompositions work better in your framework? Does your framework remove the need for curriculum learning?

Overall, I believe this is in interesting piece of work and I believe would be of interest to ICLR community.

---

> ### Author Response · Authors · 2017-12-27
> **Response to Reviewer 2**
>
> Thank you for your very valuable comments. We address your questions below.
>
> In regard to the choice of partitions: to address any potential concern regarding the partitions, we added additional experiments in Appendix D where the partitions are determined automatically, rather than being hand-specified. It is true that some care must be taken to get reasonable partitions, although our experiments suggest that even a simple K-means method can produce good results automatically. In Appendix D, we evaluate DnC on contexts generated by a K-means clustering procedure on the initial state distribution for the Picking task, which performs comparably to our manually designed contexts, indicating that performance of DnC is not particular to our choice of decomposition. We intend to extend this procedure to all the tasks for the final version. We further believe that it’s possible to find more sophisticated automatic methods to generate the decompositions, which would make for interesting future work.
>
>
> Regarding the complexity of the pairwise KL divergence, we have updated the paper to include a discussion of the computational cost in the fourth paragraph of Section 4.2.  Empirically we find that the quadratic penalty is not a bottleneck for the problems we hope to address with DnC, since sampling the environment is by far the most computationally demanding operation.
>
> In regard to the relationship with curriculum learning, we have now added some remarks at the end of the first paragraph of Section 2. Investigating the use of progressive decompositions with our method is an interesting direction for future work!

---

### Official Review · AnonReviewer3 · 2017-11-28
**Interesting approach, but seems like a fairly incremental advance on previous work**

**Rating:** 4
**Confidence:** 4

**Review:**

This paper presents a method for learning a global policy over multiple different MDPs (referred to as different "contexts", each MDP having the same dynamics and reward, but different initial state).  The basic idea is to learn a separate policy for each context, but regularized in a manner that keeps all of them relatively close to each other, and then learn a single centralized policy that merges the multiple policies via supervised learning.  The method is evaluated on several continuous state and action control tasks, and shows improvement over existing and similar approaches, notably the Distral algorithm.

I believe there are some interesting ideas presented in this paper, but in its current form I think that the delta over past work (particularly Distral) is ultimately too small to warrant publication at ICLR.  The authors should correct me if I'm wrong, but it seems as though the algorithm presented here is virtually identical to Distral except that:
1) The KL divergence term regularizes all policies together in a pairwise manner.
2) The distillation step happens episodically every R steps rather than in a pure SGD manner.
3) The authors possibly use a TRPO type objective for the standard policy gradient term, rather than REINFORCE-like approach as in Distral (this one point wasn't completely clear, as the authors mention that a "centralized DnC" is equivalent to Distral, so they may already be adapting it to the TRPO objective? some clarity on this point would be helpful).
Thus, despite better performance of the method over Distral, this doesn't necessarily seem like a substantially new algorithmic development.  And given how sensitive RL tasks are to hyperparameter selection, there needs to be some very substantial treatment of how the regularization parameters are chosen here (both for DnC and for the Distral and centralized DnC variants).  Otherwise, it honestly seems that the differences between the competing methods could be artifacts of the choice of regularization (the alpha parameter will affect just how tightly coupled the control policies actually are).

In addition to this point, the formulation of the problem setting in many cases was also somewhat unclear.  In particular, the notion of the contextual MDP is not very clear from the presentation.  The authors define a contextual MDP setting where in addition to the initial state there is an observed context to the MDP that can affect the initial state distribution (but not the transitions or reward).  It's entirely unclear to me why this additional formulation is needed, and ultimately just seems to confuse the nature of the tasks here which is much more clearly presented just as transfer learning between identical MDPs with different state distributions; and the terminology also conflicts with the (much more complex) setting of contextual decision processes (see: https://arxiv.org/abs/1610.09512).  It doesn't seem, for instance, that the final policy is context dependent (rather, it has to "infer" the context from whatever the initial state is, so effectively doesn't take the context into account at all).  Part of the reasoning seems to be to make the work seem more distinct from Distral than it really is, but I don't see why "transfer learning" and the presented contextual MDP are really all that different.

Finally, the experimental results need to be described in substantially more detail.  The choice of regularization parameters, the precise nature of the context in each setting, and the precise design of the experiments is all extremely opaque in the current presentation.  Since the methodology here is so similar to previous approaches, much more emphasis is required to better understand the (improved) empirical results in this eating.

In summary, while I do think the core ideas of this paper are interesting: whether it's better to regularize policies to a single central policy as in Distral or whether it's better to use joint regularization, whether we need two different timescales for distillation versus policy training, and what policy optimization method works best, as it is right now the algorithmic choices in the paper seem rather ad-hoc compared to Distral, and need substantially more empirical evidence.

Minor comments:
• There are several missing words/grammatical errors throughout the manuscript, e.g. on page 2 "gradient information can better estimated".

---

> ### Author Response · Authors · 2017-12-27
> **Response to Reviewer 3: Part 1**
>
> Thank you for your valuable suggestions!
>
> We have included specific experiment details in Appendix A. In particular, we ran an extensive penalty hyperparameter sweep for DnC, centralized DnC, and Distral on each task to select the appropriate parameter for each method. Since the initial version, we have also updated the experiments by conducting a finer hyperparameter sweep and by running experiments with 5 random seeds instead of 3. We have updated the paper with the results obtained from these searches (Figure 1,Table 1). We thus contend that the difference between the performance of the various methods is not contingent on the exact choice of hyperparameters, and is indeed a result of the algorithmic differences. If the reviewer has any other suggestions for how to address this concern, we would be happy to incorporate them. We have also included more comprehensive task information, which detail precisely what the contexts are in each task, in Appendix B. We have updated the paper to distinguish our use of the word “context” from contextual MDPs in Section 3. We also clarify in Section 5 that our analysis ports Distral to the TRPO objective. While the original Distral paper uses soft Q-learning, we adapt the algorithm to TRPO, since empirically TRPO exhibits better performance on high-dimensional continuous control tasks.  If the reviewer has further recommendations, we would be happy to address these as well.

---

> ### Author Response · Authors · 2017-12-27
> **Response to Reviewer 3: Part 2**
>
>
> We now address concerns regarding the differences between our method and Distral [1]. DnC and Distral not only have completely different motivations, but the technical differences between the two algorithms are substantial as well. It is worth noting that our experiments (with hyperparameter searches and multiple random seeds) over five varied tasks in the locomotion and manipulation settings clearly illustrate that the Distral method as described in prior work does not solve the challenging tasks in our evaluation, while our approach does. This extensive comparative evaluation already establishes a clear contribution over the prior work, as noted by the other two reviewers.
>
> There are also significant conceptual differences. Distral considers a transfer learning setting, while the goal in our work is to obtain a single policy that succeeds on a single challenging task with stochastic structure. While both algorithms could be applied to both settings, we feel this conceptual difference is very important. Whereas our method is concerned with the performance of the central policy on the full state space, the Distral paper evaluates performance of the local policies on their respective domains.
>
> Furthermore, Distral does not propose nor analyze the potential to solve challenging continuous control tasks with stochastic initial state distributions. The observation that decomposing the initial state distribution in this way leads to drastically improved performance is not at all obvious, and is a key insight of our work. We believe that this contribution will be highly relevant to researchers interested in solving complex continuous control tasks, and this contribution is not present in the Distral paper. In the updated paper, we also describe how to automate the process of generating these decompositions, and present results in Appendix D. We find that DnC with this automated partitioning performs comparably to the manual partitions outlined in the paper, without the need for any manual specification of partitions.
>
> Both DnC and Distral maintain the core idea that optimizing local or instance-specific policies can simplify many tasks. This idea is not new, and is popular in the RL community after works related to guided policy search [2]. In fact, ideas of the same flavor are present even in older works like target propagation [3] where an optimization method generates targets for a supervised learning network. From a bird’s-eye perspective, all these methods exploit the same principle, but a closer look at the technical details unveil significant differences.
>
> For example, GPS observes that adding a regularization term to stay close to the central network helps with distillation and overall convergence. Distral rediscovers the exact KL regularization and supervised distillation procedure as GPS, albeit with neural networks as local policies.  However, Distral’s key innovation comes from carefully choosing the algorithms for local training and applying the method to challenging visual transfer learning scenarios, something that the basic guided policy search algorithm does not do. In the same way, we propose a modified method with pairwise KL regularization terms and a varied distillation schedule, and apply it to challenging stochastic initial state continuous control tasks, as compared to the discrete control setup in Distral. Furthermore, while Distral uses soft Q-learning for their discrete action tasks, we use TRPO due to its stable performance in continuous control tasks. From this perspective, we believe that the difference between our method and Distral is comparable, if not greater than, the difference between Distral and GPS.
>
> In motivation, technical detail, and empirical performance, DnC varies significantly from Distral. Thus, we believe that the proposed method, DnC, is quite different from previous methods, a sentiment that is shared by the other two reviews as well.
>
>
> [1] Teh. et al, Distral, NIPS 2017
> [2] Levine, et al, Guided Policy Search, ICML 2013
> [3] see references of Lee et al, Difference Target Propagation, ECML PKDD 2015

---

### Public Comment · (anonymous) · 2018-02-17
**what is the x-axis of figures in Figure 1.**

Thanks for the work,  could you please tell me what is the x-axis of figures in Figure 1.

---

### Decision · Program_Chairs · 2018-01-29
**ICLR 2018 Conference Acceptance Decision**

**Decision:**

Accept (Poster)

**Comment:**

This paper proposes a specific architecture for training an ensemble of separate policies on a family of easier tasks with the goal of obtaining a single policy that can perform well on a harder task. There are significant similarities to the recently published Distral algorithm, but I am convinced that this work offers a meaningful contribution beyond that work. Moreover, the authors performed a thorough comparison between their method and Distral and found that DnC performs better.